# Broad Environmental Tolerance for a *Salicola* Host-Phage Pair Isolated from the Cargill Solar Saltworks, Newark, CA, USA

**DOI:** 10.3390/microorganisms7040106

**Published:** 2019-04-19

**Authors:** Meghan L. Rodela, Shereen Sabet, Allison Peterson, Jesse G. Dillon

**Affiliations:** Department of Biological Sciences, California State University, Long Beach, CA 90840, USA; meghan.rodela@hotmail.com (M.L.R.); shereen.sabet@csulb.edu (S.S.); allie.peterson@me.com (A.P.)

**Keywords:** *Salicola*, phage, halophile, saltern, thermotolerant

## Abstract

Phages greatly influence the ecology and evolution of their bacterial hosts; however, compared to hosts, a relatively low number of phages, especially halophilic phages, have been studied. This study describes a comparative investigation of physicochemical tolerance between a strain of the halophilic bacterium, *Salicola*, isolated from the Cargill Saltworks (Newark, CA, USA) and its associated phage. The host grew in media between pH 6–8.5, had a salinity growth optimum of 20% total salts (ranging from 10%–30%) and an upper temperature growth limit of 48 °C. The host utilized 61 of 190 substrates tested using BIOLOG Phenotype MicroArrays. The CGφ29 phage, one of only four reported *Salicola* phages, is a DNA virus of the *Siphoviridae* family. Overall, the phage tolerated a broader range of environmental conditions than its host (salinity 0–30% total salts; pH 3–9; upper thermal limit 80 °C) and is the most thermotolerant halophilic phage ever reported. This study is the most comprehensive investigation to date of a *Salicola* host–phage pair and provides novel insights into extreme environmental tolerances among bacteriophages.

## 1. Introduction

Viruses have been found to infect cells in all three domains of life and are ubiquitous and abundant with an estimated 10^31^ viruses on the planet [1,2,3]. As such, they have a great ecological and evolutionary influence on their hosts [4,5,6,7,8]. Marine host–virus systems have been relatively well studied (e.g., reviewed in [7,9,10]); but few studies exist for host–virus interactions in extreme hypersaline environments, especially for halophilic bacterial host–phage pairs [11,12,13,14]. Halophiles are organisms that grow optimally above 3.5% total salts content [15,16], while extreme halophiles display optimal growth at or above 15% total salts [17,18,19,20]. 

Solar salterns are man-made hypersaline habitats that are constructed worldwide to harvest salt, especially in coastal areas [21,22,23,24,25,26]. At the highest salinities in salterns, biological diversity is often dominated by heterotrophic prokaryotes. Many of these are culturable halophiles, including haloarchaeal lineages such as *Halorubrum*, *Haloferax*, *Haloarcula* and *Haloquadratum* [27,28,29,30], as well as bacterial genera such as *Salinibacter*, *Salicola*, and some members of the genus, *Halomonas* [30,31,32,33]. Salterns have also been shown to contain diverse communities of viruses, especially those that infect halophilic prokaryotes [34,35,36,37,38,39,40,41,42,43,44]; however, the virus populations in many salterns remain uninvestigated. Thus, they represent excellent habitats for uncovering novel haloviruses displaying extreme environmental tolerances. 

One such saltern is the Cargill Solar Saltworks (Newark, CA, USA) where relatively little research has been performed and no virus studies have been reported [45,46,47]. The purpose of this study was to perform a comparative examination of a *Salicola* host–phage pair isolated from the Cargill saltern. *Salicola* is a relatively understudied taxon of halophiles, but has recently been studied for possible biotechnological applications such as the production of a therapeutic enzyme that could potentially target cancer [48] and the production of novel hydrolytic enzymes [49]. Prior to this study, only two previous studies have described *Salicola* phages, but those did not fully characterize either the host or phage [15,50]. *Salicola* phages and other extreme hypersaline viruses have been isolated from sites around the world [50,51,52,53,54,55,56,57,58,59], but in North America, only moderately hypersaline viruses have been isolated from Mono Lake [60] and the Great Salt Lake [61]. In addition to being the first substantive report of a *Salicola* phage, this study is also the first report of any extreme hypersaline virus isolated in North America.

## 2. Materials and Methods

### 2.1. Field Sampling

Surface water samples (500 mL) were collected and environmental measurements performed from the shore of an evaporation pond (M5) of the Cargill Salt Works, Newark, CA in January 2008, during a rainfall event. Salinity, pH, and temperature were measured using a handheld refractometer (VWR, Brisbane, CA, USA) and a portable probe, respectively (Thermo Electron, Beverly, MA, USA). 

### 2.2. Host Cultivation and Isolation

Samples were processed for cultivation using sterile techniques in a research laboratory at NASA Ames Research Center (Mountain View, CA, USA) prior to transport to California State University, Long Beach. In addition, triplicate most probable number (MPN) dilution series of the M5 samples to 10^−10^ were initiated while at NASA Ames using modified growth media (MGM) [62] with 25% total salts (NaCl 3400 mM, MgCl_2_-6H_2_O 120 mM, MgSO_4_-7H_2_O 120 mM, KCl 80 mM, CaCl_2_ 4 mM). Dilutions were transported at ambient temperature and were incubated at 42 °C upon return to CSULB. Tubes were monitored over a three-week period. Growth was first observed after 96 h. Streak-plate isolations on 23% MGM agar were performed using inoculum from the highest dilution tubes that showed growth and repeated five times to ensure strain purity.

### 2.3. 16S rRNA Gene Host Identification and Phylogenetic Analysis

DNA was extracted from cell pellets from 10 ml cell suspensions centrifuged at 3220 g using a Sorvall RC-5B Refrigerated Super Speed Centrifuge and SS-34 rotor (Kendro Laboratory Products, Ridgefield, CT, USA). Cells were lysed via resuspension and vortexing in Nanopure water and nucleic acids purified via phenol extraction and ethanol precipitation [62]. Primers targeting 16S rRNA genes of Bacteria (GM3f, GM4r) [63] and Archaea (Arch 21f, 1392r) [64] were used for PCR amplifications. PCR reactions (50 μL) consisted of 1X PCR Buffer (Promega, Madison, Wisconsin), 10 mM each dNTP Mix (Promega), 1.25 mM MgCl_2_ (Promega), 20 picomoles of each primer (Eurofins MWG Operon, Huntsville, Alabama), 1.5 units GoTaq (Promega) and 10–20 ng of extracted nucleic acid. PCR reactions were set at an initial denaturation at 94 °C for five min, followed by 30 cycles of 30 s at 94 °C, 30 s at 50 °C, and 90 s at 72 °C, and a final elongation of 10 min at 72 °C. Successful amplifications were purified using a Gene Catch PCR clean up kit (Epoch Biolabs, Sugar Land, TX, USA). Samples were sent to the Macrogen Company (Seoul, South Korea) for Sanger sequencing. The sequence was deposited in Genbank under accession number KF201690.

Sequences were imported into ARB software (Version 5.2) [65] and aligned to similar sequences in the SSU REF 102 SILVA database [66]. Regions of uncertain alignment were excluded using a custom filter, yielding a 1339 bp sequence. Phylogenetic relationships were determined with the exported alignments using the neighbor-joining and parsimony methods implemented in PAUP [67]. Bootstrap confidence values were obtained via 1000 pseudoreplicates for each. The isolate of interest in this study, hereafter CGM5-S, was the only isolate belonging to the bacterial genus *Salicola.*


### 2.4. Physicochemical Tolerance Experiments

The CGM5-S host was characterized by determining optimal growth ranges for salinity (total salts), temperature, and pH. Sterilized culture bottles were used to inoculate 300 µL of exponential phase cells into 20 ml of each media treatment in triplicate. At each time point, 300 µL was sampled without replacement from each culture to measure the optical density at 600 nm (OD_600_) in a Genesys 10 UV spectrophotometer (Thermo Scientific, Waltham, MA, USA).

For salinity experiments, MGM was made with an increase in total salts of 5% salinity increments (5%–30%) and the pH was adjusted to 7.5 using Tris Base [62]. Cultures were incubated at 42 °C and sampled every 6 h for 48 h. For pH experiments, 20% MGM (the optimal growth salinity) was used for all treatments. The pH of the medium was adjusted at half unit increments (5–8.5) using Tris Base or for more extreme pH values HCl (1 N) or NaOH (1 N) and then buffered with the appropriate Good’s buffers (e.g., MES (15 mM) and MOPS (2.4–9.6 mM)), to obtain the desired pH. For each pH treatment, a fourth culture replicate was grown along with the three experimental cultures. This culture was periodically subsampled (1 mL) to monitor pH changes over the course of the experiment, and parallel adjustments were made to all four experimental bottles using 1 N HCl and 1 N NaOH as necessary without contaminating the experimental bottles. pH was confirmed in all bottles at the end of the experiment. Cultures were incubated at 42 °C and 1 ml samples were taken every 8 h for 48 h. Temperature experiments were performed using 20% MGM adjusted to pH = 7.5 at 32, 39.5, 43, 45.5, 49, 50.5, 55, and 57 °C. Unlike salinity and pH experiments that were run in parallel, temperature experiments were done serially due to a limited number of incubators. Samples were taken every eight hours for 48 h. 

For all experiments, OD_600_ values were measured spectrophotometrically to calculate cell counts. OD_600_ was converted to colony forming units (CFU) for further data analysis using a formula derived empirically from plate counts. A linear regression analysis performed using StatView 5.0.1 (SAS Institute Inc., Cary, NC, USA) comparing counts and OD_600_ resulted in the following formula: CFU mL^−1^ = −15106624.773 + (1211807898.255 × OD_600_).(1)

From cell count data, μMax values were determined from the interval with the highest growth rate (μ) as previously described [30]. 

### 2.5. Microscopic and Biochemical Host Characterization

Cells were Gram stained using the approach of Dussault [68] that uses fixation in 2% acetic acid instead of heat. Gram stained hosts were imaged using an IX81 inverted scope at 100x (Olympus, Center Valley, PA, USA) and C4742-95 digital camera (Hamamatsu, Bridgewater, NJ, USA). Average cell size (n = 20) was calculated for cells measured using Slide book software v.5.0.0.1 (Intelligent Imaging Innovations, Denver, Colorado, USA). Catalase and oxidase tests were performed using standard reagents according to manufacturer’s directions (Becton, Dickerson, and Co., Sparks, Maryland, USA). To characterize growth on a range of carbon substrates, cells were washed in triplicate, resuspended in 20% salt water (total salts), and assayed using the PM1 and PM2A BIOLOG Phenotype MicroArrays (BIOLOG, Hayward, CA, USA) as previously described in detail by Sabet et al. [30]. 

### 2.6. Phage Isolation and Enumeration

Viruses were isolated from 1 ml of M5 pond water centrifuged at 9300 g (5415D, Brinkmann Instruments, Westbury, NY, USA) for 5 min to pellet the larger organisms and debris. The supernatant was filtered using a 0.2 µm filter (EMD Millipore, Billerica, MA, USA) and the filtrate was mixed with 300 μL of exponential phase host cells and used to perform a top agar overlay [69]. The plates were incubated at 42 °C until plaque formation (~1 week). Plaques were sterile-picked and suspended in 300 μL of 23% total salts solution for a minimum of 24 h. This sample was serially diluted to 10^−6^ and the dilution was used to do another round of top agar overlay and plaque isolation. This process was performed five times to ensure strain purity of the viral stock [62]. The sole virus, designated as CGφ29, that infected the *Salicola* host was selected for characterization in this study.

Over the course of a number of experiments described below, top agar overlay assays were used to measure the titer of the phage (PFU mL^−1^). After treatment under various conditions (i.e., salinity, pH and temperature), 3 mL of top agar was mixed with 100 µL of phage and 300 µL of host cells (grown under optimal conditions), then poured evenly onto plates in triplicate. Inoculated plates were allowed to solidify and then incubated at 42 °C for 72 h and observed for plaque formation. Average PFU mL^−1^ was calculated for triplicate plates accounting for dilution. 

### 2.7. Generating Large-Scale Viral Stocks

Initial viral stocks yielded low titers (10^8^–10^9^ PFU mL^−1^). To achieve stocks with high enough titer, we used a large-volume cultivation method. Host cells were grown to OD_600_ 0.7 and then infected in a total volume of 800 mL with stock virus at a MOI of 1, which produced lysates of 10^12^ PFU mL^−1^. Cultures were monitored for 48 h until evidence of lysis was apparent by a decrease in turbidity and the presence of cell debris. The total lysate volume was collected and transferred to 500 mL centrifugation bottles and the cell debris was pelleted using a JLA 10.5 rotor in an Avanti J-E centrifuge (Beckman, Indianapolis, Indiana, USA) at 18,592 *g* for a total of 50 min at 4 °C. The supernatant was concentrated using centrifugal filter units (Centricon Plus-70, EMD Millipore) according to the manufacturer’s instructions and with the same centrifuge but with a JS 5.3 rotor (Beckman) at 3500 *g*. To further concentrate phage particles, the concentrated lysate was ultracentrifuged in an Optima L-100XP using the SW41 Ti rotor (Beckman) at 274,355 *g* at 4 °C for 90 min. Most of the supernatant was removed and the pellet resuspended in 1 ml of supernatant. The concentrated virus was incubated for 48 h at 4 °C. Viral titer (PFU mL^−1^) was determined using the top agar overlay described above.

### 2.8. CGφ29 Genome

Phage DNA was obtained from viral stocks concentrated in the manner described in the previous section. To eliminate any external or foreign nucleic acids, phage particles were suspended in 10 mL of phage resuspension solution [62], then 20 U mL^−1^ of Optimize Recombinant DNase I (Fisher Scientific, Pittsburg, PA, USA) and 10–20 U mL^−1^ RNase One (Promega) were added with buffer according to manufacturers’ instructions and incubated at 37 °C for 1.5 h. To deactivate the DNase I and RNase enzymes, proteinase K (50 μg mL^−1^) (EMD Chemicals, Gibbstown, NJ, USA) was added and the sample was incubated for 10 min at 37 °C. To denature the phage capsid and release the nucleic acids, one fourth volume of proteinase K buffer (5X) was added and the solution was heated to 80 °C for 10 min, then cooled to room temperature; 100 μg mL^−1^ proteinase K was added and the mixture was incubated at 50 °C for 2 h [62]. Nucleic acids were precipitated using the same procedure as described above for the host DNA. 

Purified nucleic acids were treated with 40 U mL^−1^ Optimize Recombinant DNase I (Fisher Scientific) or 20 U mL^−1^ RNase One (Promega). Treated nucleic acid samples were electrophoresed on a 1% agarose gel, stained with ethidium bromide (0.5 µg mL^−1^) and visualized using UV transillumination along with untreated nucleic acid and λ DNA/ Hind III markers (Promega) as references.

The genome was sequenced by the Broad Institute as part of the Gordon and Betty Moore Marine Phage Sequencing project [70]. The sequence and annotation are available in Genbank under accession HQ634153. Additional annotation and genome architecture analyses were performed using the DOE JGI IMG [71] and is available under JGI IMG Genome 2588253651. The large subunit terminase sequence from CGφ29 was compared using blastp at NCBI and related sequences downloaded from Genbank and imported into Geneious Prime software (v. 2019.04) where they were aligned using the MUSCLE aligner function and a neighbor-joining phylogeny constructed using the Tree Builder function. Bootstrap confidence values were obtained via 1000 pseudoreplicates.

### 2.9. TEM of CGφ29 Phage

CGφ29 was imaged using a JEM–1200 EXII Transmission Electron Microscope (JEOL, Tokyo, Japan) and a C4742-95 digital camera (Hamamatsu). Five μL of phage suspension was adsorbed to a 400 mesh Formvar copper film grid (Electron Microscopy Sciences, Hatfield, PA, USA) for 2 min. The grid was rinsed with sterile Nanopure water for 15 s and stained with 1% *w*/*v* uranyl acetate (Ted Pella, Redding, CA, USA) for 2 min, air dried, and imaged. 

### 2.10. CGφ29 Host Range 

The host range of CGφ29 phage was tested by spot plating 5 µL of phage stock onto solidified host top agar overlays, incubating overnight at 42 °C, and observing for plaque formation. The *Salicola* phage was tested against six *Salicola* isolates (9-A-U, 11-A-U, 12-A-U, 11GM-A-U, 11GM-B-U, and CH-A-U) previously isolated in our laboratory from the Exportadora de Sal (ESSA) saltern in Guerrero Negro, Baja California, Mexico [30] as well as the six strains isolated from the Maras saltern in Peru, including the type strain, *S. marasensis* [25].

### 2.11. Chloroform Sensitivity and Environmental Tolerance of CGφ29 Phage

Phage sensitivity to chloroform was tested in triplicate using a modified version of the protocol described in Chow and Rouf (1983). CGφ29 (PFU mL^−1^ = 2.54 × 10^12^) was diluted 1:100 in 1 mL of 20% total salts solution containing 0.05% chloroform (Sigma, St. Louis, MO, USA). A positive control was set up with phage diluted into 20% total salts solution without chloroform, and a negative control contained 0.05% chloroform but no phage. Each treatment was inverted by hand for 10 min and centrifuged at 9300 *g* for 5 min [72] (5415D, Eppendorf) and viral titer (PFU mL^−1^) calculated using the top agar overlay as described above. 

CGφ29 was examined for sensitivity to varying salinity (one-year exposure), pH (72 h time-course exposure), and temperature (72 h time-course exposure). For these experiments, CGφ29 stock (PFU mL^−1^ = 2.54 × 10^12^) was diluted 1:100 into each respective treatment. For salinity experiments, phage stock was diluted into 0–30% total salts solutions and incubated at room temperature for one year. For pH tolerance, the phage was diluted into 20% total salts solution at varying pH treatments (pH 2–9) created using buffers as described above and incubated at 42 °C. For temperature experiments, CGφ29 was diluted into 20% total salts solution at a pH of 7.5 and incubated in a water bath at 65, 70, 75, and 80 °C and in an incubator at 42 °C. In each experiment, all treatments were performed in triplicate. Unlike the salinity samples, which were sampled once, samples for both pH and temperature were taken over a time course at 1, 4, 8, 24, 48, and 72 h of exposure. Top agar overlay assays were performed to calculate PFU mL^−1^ for treated phage.

### 2.12. One-Step Growth Curve

Exponential-phase host cells were inoculated with virus at MOI 30 and incubated with shaking (150 rpm) at 42 °C for 2 h for adsorption. Cells were then washed of unadsorbed virus three times using 20% total salts solution and resuspended in 10 mL of 20% MGM medium. An uninfected control culture was processed in the same manner without phage addition. Experimental and control cultures were incubated at 42 °C and sampled every 2 h for 30 h to perform cell counts (CFU mL^−1^; experimental and control) and viral titers (PFU mL^−1^; experimental only).

### 2.13. Statistical Analysis

Statistical analyses were performed using StatView 5.0.1 (SAS Institute Inc.). An unpaired t-test was used to analyze CGφ29 chloroform sensitivity. A one-factor analysis of variance (ANOVA) was used to analyze the response of CGφ29 to varying salinities. Repeated measures analysis of variance (RM-ANOVA) were used to measure CGφ29 response to pH and temperature as well as the response of the *Salicola* host to salinity, pH, and temperature. For the salinity response experiment, growth at 10% total salts was highly variable and these data were removed from statistical comparisons since they violated the assumptions of equal variances of the ANOVA. Fisher’s PLSD pairwise post-hoc tests were applied to all ANOVAs. 

## 3. Results and Discussion

During collection, Pond M5 had a total salinity of 25%, pH of 7.7, and temperature of 9.8 °C. The MPN estimation of the cultivable cell density in pond M5 was 2.4 × 10^8^ CFU mL^−1^. Initially, one *Halorubrum*, three *Halomonas*, one *Halovibrio*, and one *Salicola* species were obtained from streak-plate isolations of the positive MPN tubes as identified by 16S rRNA sequencing. The focus of this study was to compare environmental tolerances of novel host–phage pairs, and subsequently all isolates were screened for viruses. The strain CGM5-S and its phage, CGφ29, were selected because no previous study had described a *Salicola* host–phage pair.

### 3.1. Host Characterization

The *Salicola* isolate, that is the focus of this study (CGM5-S; Figure 1), was the only Cargill bacterial isolate from which a phage was successfully isolated. The genus *Salicola* (Class Gammaproteobacteria) was first identified from six unique isolates from the Maras solar salterns in Peru, including the type species *Salicola marasensis* [25]. *Salicola* strains have also been isolated from hypersaline lakes, sabkhas, and salterns in Spain, Algeria, Iran, and Mexico [30,32,73,74,75].

Similar to described strains from these sites, CGM5-S cells are Gram-negative rods with an average cell length of 1.6 μm. CGM5-S is catalase positive and oxidase negative, in contrast to the type strain *S. marasensis* and *S. salis*, which are both catalase and oxidase positive [25,74]; however, variation in oxidative enzyme content in this group has been observed, as other isolates that are catalase and oxidase negative have been reported [73]. CGM5-S is identical in its 16S rRNA gene sequence to *Salicola* strain PV-3, isolated from an Italian solar saltern [15], and a *Salicola* strain isolated from the Exportadora de Sal (ESSA) saltern in Baja California [30] (Figure 1) suggesting that this specific *Salicola* phylotype may be a cosmopolitan saltern specialist. 

*Salicola* str. CGM5-S utilized 61 of 190 (32%) of the carbon sources tested, including eight sugars and 8/17 (47%) of the amino acids included in the BIOLOG plates (Appendix A). It is impossible to fully compare substrate usage profiles for our *Salicola* isolate with most other strains due to limited overlap of tested substrates as many studies have not used comprehensive assays such as BIOLOG, an approach only recently developed for use with extreme halophiles [30]. Indeed, the best comparison we could make was with *Salicola* strain 9-A-U isolated from the ESSA saltern in that study (Appendix A). Both the Cargill and ESSA saltern *Salicola* strains appeared to be substrate generalists, capable of growth on many different carbon sources including 10 common substrates. Other halophilic genera, especially haloarchaeal genera, have been shown to be more selective in their substrate usage patterns [30]. It is unknown what the availability of carbon substrates is in the Cargill pond from which this strain was isolated; however, the ability to use a broader range of substrates for growth may allow halophilic bacteria to compete with halorarchaea, which are often numerically dominant in salterns [76]. Further study of carbon substrate utilization in this and other halophilic genera and substrate availability in hypersaline salterns is needed.

### 3.2. Phage CGφ29 Characterization

Phage CGφ29 is a head/tail virus, which has a long, non-contractile tail that lacks visible tail fibers indicating it is a member of the *Siphoviridae* family (Figure 2a). Previously described phages that were isolated from *Salicola* have been classified as members of the *Siphoviridae* or *Myoviridae* families [15,50]. CGφ29 had an average tail length of 179 nm and a head diameter of 51 nm. Figure 2b shows CGφ29 particles adsorbing to the host cell, with the long, flexible tails clearly visible. Nucleic acid typing indicated that CGφ29 is a DNA virus as evidenced by the complete digestion of the phage nucleic acid when treated with DNase and no evidence of digestion when treated with RNase (Figure 2c). The circular genome was estimated to be approximately 40.7 kb with 37.3 kb of coding bases and a guanine + cytosine (GC) content of 56.4%. Sixty-four putative open reading frames were found in the genome sequence, 17 (26%) of which were identifiable via similarity with amino acid sequences in the JGI IMG/Virus database including two helicases and single copies of terminase, peptidase, transposase and DNA methylase sequences (Figure 3, Appendix A). The relatively low level of annotation is likely due to the novelty of the genome and lack of database information for environmental viruses such as CGφ29. The majority of genes are in the positive direction, but a large contiguous portion was in the negative direction. Individual genes showed variation in GC content along the genome ranging from 45%–62% (Figure 3b). NCBI Blastp results using the large subunit terminase protein sequence from CGφ29 revealed high sequence similarity (E scores 0.0, similarity 75%–79%) with terminase genes from phage infecting a number of hosts which was confirmed by placement in the phylogeny (Figure 3c). Interestingly, all hosts belonged to either the Gammaproteobacteria or Betaproteobacteria lineages. Terminase genes play an essential role in packaging in DNA viruses of the Caudovirales and as such can be used as a shared phylogenetic marker including in halophilic bacteriophage [77].

Despite the close phylogenetic relationship between CGM5-S and the ESSA *Salicola* hosts (Figure 1), CGφ29 was unable to infect any of the ESSA *Salicola* isolates in plaque assays, even the isolate with identical 16S rRNA genotype (data not shown). The phage was also unable to infect any of the six *Salicola* strains originally isolated from the Maras saltern in Peru, including the type strain *S. marasensis* [25]. This suggests that CGφ29 may have high specificity for the CGM5-S host. Host specificity has also been observed in a recent study of *Salicola* phage SCTP-3 [15]. However, another phage in that study, SCTP-2, was able to infect two hosts, indicating that variation in host specificity occurs. These differences in host specificity have been observed in haloarchaeal viruses as well [15,78].

For generating our large-scale viral stocks, we utilized MOI 1 in order to produce the greatest possible number of viruses from a low titer viral stock, as these infections did not need to be synchronized. However, for the one-step growth curve, MOI 1 did not produce any appreciable killing after a 1 h adsorption step; therefore, we evaluated higher MOIs between 10 and 50 to determine the optimal MOI to synchronize the CGφ29 host cells for infection. Other groups investigating haloviruses have described the use of different MOIs for different purposes within the same study. For example lower MOIs (0.005–40) have been used for lysate production while higher MOIs (3–100) have been employed for one-step growth experiments [51,54,78,79,80,81]. In our evaluation, all higher MOIs (10–50) resulted in an average killing of 88% after 2 h adsorption, which is very similar to that discovered for the Hs-1 virus [82]. Therefore, we used MOI 30 to synchronize CGM5-S host cells for the viral lifecycle experiment. 

During the one-step growth curve of CGφ29, the virus titer sharply increased between 6 and 10 h after infection (Figure 4a) and the burst size was calculated to be 5646. Infected cell cultures continued to grow but did not reach the same densities as uninfected cultures, resulting in a widening difference in cell densities between infected and uninfected cultures over the course of the experiment, but not complete cell death (Figure 4b). This is consistent with our observation that CGφ29 plaques are often turbid with resistant cells growing in the middle of the plaques (Figure 2d; Michael Dyall-Smith, personal communication) and indicates that either the virus is lytic and some host cells in the population are naturally resistant to infection, or the virus may be temperate. Our findings correspond with direct visual observations of the liquid infection cultures, which showed the presence of lytic cell debris, but the cultures never fully cleared. Although many halovirus hosts have shown a more dramatic lysis [14,78], others have shown infection dynamics similar to CGφ29 in which the host cells of the infected culture continued to grow [82,83].

### 3.3. Host and Phage Are Both Broadly Tolerant of Environmental Conditions

As expected, significant effects of salinity, pH, and temperature on the growth of *Salicola* str. CGM5-S over time were observed (Figure 5; RM-ANOVAs, *p* < 0.0001 for all three parameters). Host cells grew best at a salinity of 20% total salts and showed consistent growth between 15% and 30% total salts (Figure 5a). Growth at 10% salinity was more variable, likely due to the cells being near the lower limit of their growth tolerance range. These findings confirm that CGM5-S, like other *Salicola* strains, is among the more halophilic species in the domain Bacteria (Table 1), although it is not as halophilic as members of the genus *Salinibacter* [31].

CGM5-S was able to grow between pH 6–8.5, the highest pH tested. Although maximal growth rates varied across this range of pH, no statistically significant difference in µMax was observed (Fisher’s PLSD test, *p* = 0.32724, Figure 5b). There was no growth at pH 5 or 5.5, which explains the significant overall RM-ANOVA treatment effect noted above. This response to pH is consistent with other *Salicola* strains; only one strain has been reported to tolerate pH lower than 6 (Table 1). The Cargill *Salicola* showed optimal growth at ~41 °C and rapid growth between 32–48 °C (Figure 5c). Slower growth was observed at lower temperatures (e.g. room temperature) and no growth was observed at 50 °C (data not shown). Again this is consistent with other *Salicola* species that typically show growth optima between 35–40 °C and ranges of 20–48 °C. CGM5-S shows a thermal growth optimum close to that of the IC10 isolate [32], but an overall range more similar to *S. salis* (Table 1). 

The temperature growth rate optimum of these halophilic bacteria is on the high end of the normal environmental range for mesophilic microorganisms [16], but well below the 50 °C level, a common cut-off for defining thermophily [84]. Isolation of a strain with moderate thermal tolerance may be in part a result of selection due to the elevated temperature (i.e., 42 °C) at which this strain was cultured (as is commonly done with halophiles to speed growth [62]), but the presence of this strain in the saltern may also reflect adaptation to the fluctuating conditions of this habitat. As evaporation occurs, the decrease in water levels often leads to variable conditions, especially salinity and temperature [85].

Overall, the CGφ29 phage was more tolerant of environmental conditions than its host and was more tolerant than most other reported phages infecting halophilic bacterial hosts (Table 2). The highest average phage titer was found at 10% total salts; however, no significant differences in infectivity were observed for CGφ29 even after a one-year exposure to the salinities tested (1-way ANOVA, *p* = 0.0896) (Figure 5d). Continued infectivity after exposure to a wide range of salinities (0–30%) is similar to observations for phages from other salterns (SCTP-1, SCTP-2) and from a hypersaline soil phage (F9-11), but much greater than bacteriophage φD-86 isolated from soy sauce (Table 2). To our knowledge, the one year duration of our virus salinity treatment represents the longest reported time frame of study to date and indicates that viral particles have the potential to retain infectivity for years between host infection, a trait that may be especially important in evaporative habitats where host availability may vary widely as conditions change.

For our pH and temperature phage experiments, a shorter time of incubation was used (72 h); however, similar broad tolerances were observed in these experiments as well. The CGφ29 phage was tolerant of both acidic and basic conditions as no difference in infectivity was observed between pH 4–9 (*p* > 0.05 for all pair-wise comparisons). A significant overall pH treatment effect (RM-AMOVA, *p* ≤ 0.0001) was due to significantly decreased infectivity at pH 3 compared with all other pH values tested (Figure 5e); no plaques were observed at pH 2 (data not shown). This broad pH tolerance range is comparable to other haloviruses, both those infecting bacteria and archaea, although this is the first report of pH tolerance for a *Salicola* phage (Table 2).

Perhaps the most surprising finding of our study was the very high thermal tolerance of the CGφ29 virus (Figure 5f). The significant overall effect of temperature on infectivity (RM-ANOVA, *p* ≤ 0.0001) was due to a significant decrease in infectivity, but only above 70 °C (*p* < 0.05 for all pairwise comparisons with lower temperature treatments). Remarkably, despite a decrease in infectivity, the Cargill *Salicola* phage could still infect its host after 8 h of exposure to 80 °C. The ability of the CGφ29 to remain infectious after prolonged exposure to temperatures up to 75 °C shows that it is extremely thermotolerant [84]. In contrast with the salinity and pH tolerances described above, the broad range of thermal tolerance of CGφ29 has not been commonly reported for haloviruses infecting bacteria or archaea. Unfortunately, thermal tolerance was not tested in the other *Salicola* phage reports, so we cannot say whether these findings are unique to this virus or a more general characteristic of the genus. In comparison with other haloviruses tested for temperature tolerance, CGφ29 tolerated temperatures as much as 25 °C above other reports and for considerably longer time periods (Table 2). To our knowledge, these findings indicate that CGφ29 is the most thermotolerant of any hypersaline virus characterized to date.

There was no significant difference observed in infectivity after exposure to chloroform compared to control (unpaired t-test, *p* = 0.9640), suggesting that CGφ29 does not contain lipids (data not shown). Chloroform sensitivity assays have not been reported for other phages infecting *Salicola* or other halophilic bacteria; however, our finding is similar to the archaeal halovirus φH that was chloroform insensitive [86]. 

Overall, the greater tolerance of the CGφ29 virus compared to its host is likely due to the life cycle of the virus. When viruses are outside their host in the environment as virion particles, a condition mimicked in these tolerance experiments, they are inactive and have no repair mechanisms to utilize if they are damaged. Phage must remain intact for potentially prolonged periods of time to be infective, so this virus may have evolved enhanced environmental tolerance via mutations in its capsid proteins or other adaptations similar to those that have been found to be associated with thermal selection [87]. Although it has been generally reported that virions have a relatively short half-life in marine habitats (on the order of ~48 h) [8], our findings, especially the year-long salinity experiment, suggest that longer survival periods in the virion stage may be possible. This would support the Bank hypothesis that suggests that phage life cycles may include extended periods of inactivity [88]. At this point, the specific mechanisms behind the high environmental tolerance of CGφ29 are unknown, although we hypothesize that the biochemical nature of the viral capsid may be responsible. Further study is required to determine if the broad salinity and extreme temperature tolerance of CGφ29 is unique to this phage or is a characteristic of *Salicola* phages and other haloviruses.

## Figures and Tables

**Figure 1 microorganisms-07-00106-f001:**
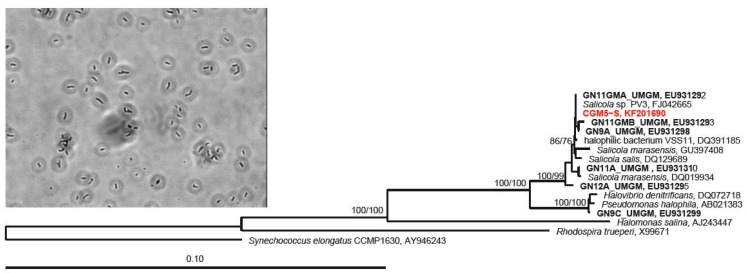
Neighbor-joining dendrogram depicting relationship of the CGM5-S host 16S rRNA gene sequence (red) with closely related sequences, and a photomicrograph of CGM5-S (inset). Guerrero Negro, Baja CA S., MX ESSA saltern strains have a GN prefix and are in bold. Numbers at nodes represent bootstrap confidence values for neighbor-joining/parsimony trees expressed as percentages from 1000 pseudoreplicates each. The Alphaproteobacterium *Rhodospira trueperi* and the cyanobacterium, *Synechococcus elongatus*, were used as outgroups.

**Figure 2 microorganisms-07-00106-f002:**
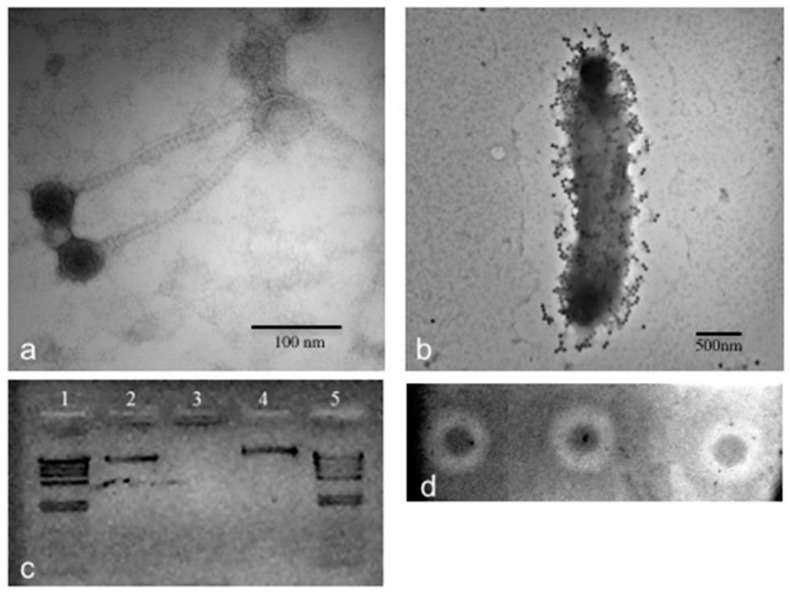
Microscopic, genomic, and phenotypic characterization of CGφ29 phage. (**a**) TEM image of CGφ29. Image was taken at 300,000× direct magnification. (**b**) TEM image of CGφ29 phage infecting its *Salicola* host. Image was taken at 30,000× direct magnification. (**c**) Nucleic acid typing for CGφ29. Lanes 1 and 5 contain λ phage control DNA cut with Hind-III. Lane 2 is uncut CGφ29 nucleic acid, lane 3 is CGφ29 nucleic acid treated with DNase and lane 4 is CGφ29 nucleic acid treated with RNase. (**d**) Spot assays of CGφ29 on a lawn of CGM5-S host cells. The inverted and contrast-enhanced image shows the plaques as white circles, or halos, with unlysed host cells as dark spots in the middle.

**Figure 3 microorganisms-07-00106-f003:**
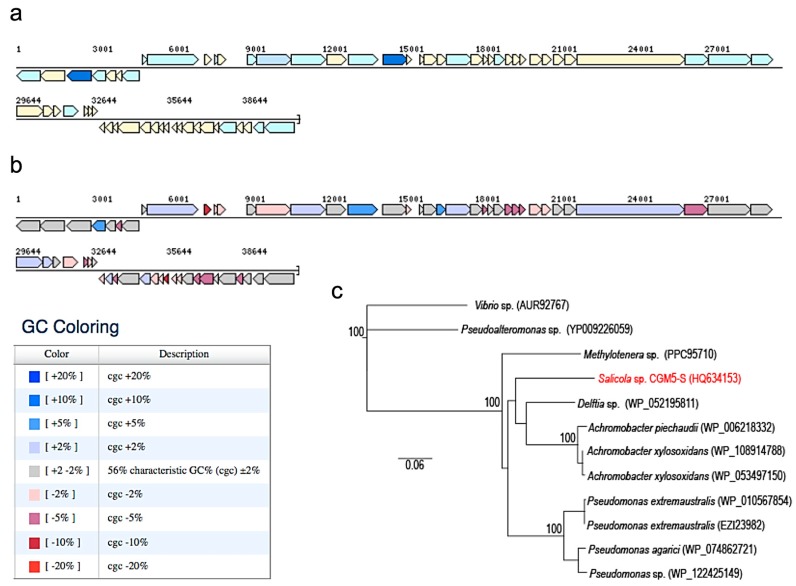
Genomic structure of CGφ29 generated using the chromosome viewer function in JGI-IMG. (**a**) Coding sequences color-coded using pFAM gene category annotations. Dark blue are ‘replication, recombination and repair’, light blue are ‘unclassified function’, and beige are ‘hypothetical proteins’. (**b**) GC content mapping with legend showing deviations from mean GC (56%). (**c**) Neighbor-joining dendrogram depicting relationship of the CGφ29 terminase gene sequence (red) with closely related sequences (species names of bacterial phage hosts are depicted on branches with accession numbers). Numbers at nodes represent bootstrap confidence values for neighbor-joining tree expressed as percentages from 1000 pseudoreplicates each. The *Vibrio* sp. and *Pseudoalteromonas* phage sequences were used as outgroups.

**Figure 4 microorganisms-07-00106-f004:**
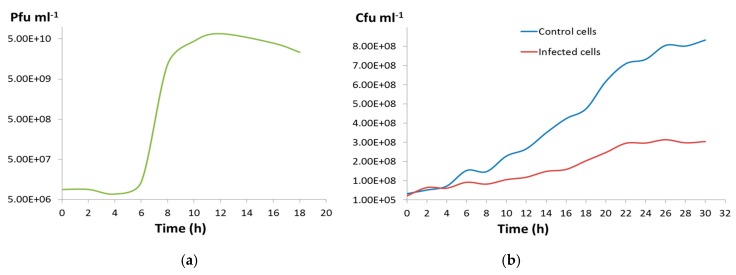
(**a**) One-step growth curve for CGφ29. (**b**) Blue line shows cell numbers per ml of an uninfected CGM5-S host culture, and red line shows mean cell numbers per ml of triplicate infected CGM5-S host cultures.

**Figure 5 microorganisms-07-00106-f005:**
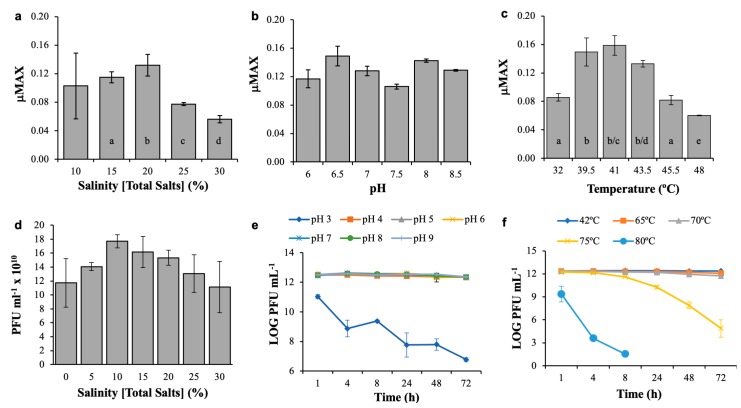
Physicochemical tolerance experiments for *Salicola* str. CGM5-S host and CGφ29 phage. Panels a, b, and c show maximal growth rates (µMax) during 48 h salinity, pH, and temperature exposure experiments for the CGM5-S host. Panels d, e, and f show CGφ29 phage titer following exposure to salinity treatments for one year, and pH and temperature treatments over a 72 h time-course, respectively. Bar graphs represent non time-course experiments; line graphs represent time-course experiments. Values are means ± Standard Deviation (n = 3). Different letters on bars denote statistically significant differences (*p* < 0.05) among treatments.

**Table 1 microorganisms-07-00106-t001:** Comparison of physicochemical responses of CGM5-S and other described *Salicola*
^1^.

Strains	CGM5-S (USA)	9-A-U (Mexico)	*S. marasensis* and Related Isolates (Peru)	*S. salis* (Algeria)	7SPE Isolates (Tunisia)	TBZ Isolates (Iran)	IC10 (Spain)
Salinity (%)							
Optimum	20	15	15	15–20	NR	2/3 grew at 20	20–25
Range	15–30	10–30	10–30	10–25	5–25	NR	15–30
pH							
Optimum	6.5	NR	7	7–7.5	NR	2/3 grew at 9	8
Range	6–8.5	NR	6–8	6–9	6–7.5	NR	5–8.5
Temperature (˚C)							
Optimum	41	NR	35	37	37	NR	40
Range	32–48	NR	20–37	30–45	NR	NR	28–40

^1^ CGM5-S (this study); 9-A-U [30]; *S. marasensis* [25]; *S. salis* [74]; 7SPE3’020, 7SPE326, 7SPE304, 7SPE3’09, 7SPE426, 7SPE3216 [73]; TBZ6, TBZ24, TBZ39 [75]; IC10 [32]. Countries of isolation in parenthesis below the isolate name. NR = not reported.

**Table 2 microorganisms-07-00106-t002:** Physicochemical responses of CGϕ29 and other described bacterial halophilic viruses ^1^.

Virus	CGϕ29	SCTP-1	SCTP-2	ϕD-86	F9-11
Source	Cargill Saltern (USA)	Saltern (Italy)	Saltern (Italy)	Soy sauce	Hypersaline soil (Spain)
Host	*Salicola* sp. CGM5-S	*Salicola* sp. PV3	*Salicola* sp. PV3	*Pedicoccus* *halophilus*	*Deleya* *halophila*
*Salinity (%)*					
Tolerance range	0–30	0–26	0–26	0–15	0–30
Maximum exposure time	1 y	15 min	18 min	1 h	45 d
*pH*					
Tolerance range	3-9	NR	NR	4.5-10.7	NR
Maximum exposure time	72 h	NR	NR	1 h	NR
*Temperature (˚C)*					
Maximum temperature tolerated	75	NR	NR	50	NR
Maximum exposure time	72 h	NR	NR	1 h	NR

^1^ CGφ29 (this study); SCTP-1, SCTP-2 [50]; ϕD-86 [13]; F9-11 [11]. NR = Not Reported.

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
