# Peer review of "Broad Environmental Tolerance for a Salicola Host-Phage Pair Isolated from the Cargill Solar Saltworks, Newark, CA, USA"

_microorganisms, 2019, doi:10.3390/microorganisms7040106_

Round 1
Reviewer 1 Report
This manuscript by Rodela et al. examines the environmental tolerance of a phage and its halophilic bacterial host, isolated from a relatively unstudied solar saltern. This article will be a valuable contribution to the field of halophilic microorganisms and their viruses, which, as the authors correctly state, are understudied. I would therefore expect this article to generate interest within its field, and to be well cited in the future.
In this manuscript, the authors have thoroughly characterized the phage and its host organism using a variety of methods in a comprehensive study, and have shown some novel observations into the environmental tolerance of phages. It is well-written and clear, and I cannot identify much that I feel would require modification. I therefore recommend this manuscript for publication almost as is, aside from the very minor points below.
Line 88: “300μL of cells” was used as an inoculum. This is very vague. Please clarify, was this 300 μL of a stationary/ exponential phase/ adjusted OD culture/ etc.? (same for line 126).
Throughout: standard convention for SI units is to have a space between the number and the unit.
Author Response
1. Line 88: “300μL of cells” was used as an inoculum. This is very vague. Please clarify, was this 300 μL of a stationary/ exponential phase/ adjusted OD culture/ etc.? (same for line 126).
> We have added “exponential phase” for clarification as requested (for both lines 88 and
126).
2. Throughout: standard convention for SI units is to have a space between the number and the unit.
>Corrected.
Reviewer 2 Report
The paper describes the environmental tolerance for a Salicola host-phage pair isolated from the Cargill Solar Saltworks
The article is very interesting being the first time that virus of Salicola strains are reported.
Line 36: Species of Halomonas are not isolated with much frequency at high salinities, they are much more frequent at intermediate salinities.
Line 61: Could you indicate the composition of the total salts?
Line 121: What culture medium have you used to inoculate BIOLOG plates? Since this isolate needs high concentrations of salt, have you added salt to the medium? Has not it interfered in the results since, being littele volume, evaporation can precipitate the salts?
It is not clear what is the initial goal of this work, I understand that from the sampling of M5 many isolates of bacteria and archaea are isolated? Why not indicate the number of isolates and those that correspond to archaea and bacteria? The work would be much more complete, although I understand that this is not the objective of it.
And of all these isolates, CGM5-S is selected for its study, is that? It is no clear, perhaps it should be clarified in the text.
In order to make a good comparative table of the use of substrates, at least the two species described Salicola marasensis and Salicola salis should be used in this study and carry out the growth in BIOLOG plates. In that case the results should be comparable because they are obtained in the same laboratory conditions.
Author Response
1. Line 36: Species of Halomonas are not isolated with much frequency at high salinities, they are much more frequent at intermediate salinities.
> Previous reports do indicate that Halomonas can be found at higher salinities e.g. (Kindzierski et al., 2017). However, we have rewritten line 36 to clarify that only some members of Halomonas are halophilic and added this reference
2. Line 61: Could you indicate the composition of the total salts?
> We have now added the total salts used for 25% MGM media.
3. Line 121: What culture medium have you used to inoculate BIOLOG plates? Since this isolate needs high concentrations of salt, have you added salt to the medium? Has not it interfered in the results since, being littele volume, evaporation can precipitate the salts?
>We clarified the procedure briefly here, but it is described in further detail in the indicated reference.
4. It is not clear what is the initial goal of this work, I understand that from the sampling of M5 many isolates of bacteria and archaea are isolated? Why not indicate the number of isolates and those that correspond to archaea and bacteria? The work would be much more complete, although I understand that this is not the objective of it.
>We added information in the results section regarding the number and genera of the
isolates.
And of all these isolates, CGM5-S is selected for its study, is that? It is no clear, perhaps it should be clarified in the text.
>We have now clarified the reasoning for choosing the Salicola host-phage pair.
5. In order to make a good comparative table of the use of substrates, at least the two species described Salicola marasensis and Salicola salis should be used in this study and carry out the growth in BIOLOG plates. In that case the results should be comparable because they are obtained in the same laboratory conditions.
> We agree that these data would have particular merit in a comparative study. However, the focus of this study was primarily to characterize only the host-phage pair that we isolated. The other Salicola species were used to test host range of the phage, but we felt that further comparison of BIOLOG was beyond the scope of this study and would be better reserved for a future study.
Round 2
Reviewer 2 Report
Thank you very much for taking into account my suggestions.
I have read carefully all the comments made by the authors answering my questions and their answers have been clear and precise.
I think that with these comments the article has improved and for my part, it is correct